# In situ visualization of opioid and cannabinoid drug effects using phosphosite-specific GPCR antibodies

Sebastian Fritzwanker[1], Falko Nagel[2], Andrea Kliewer [1], Viviane Stammer[1] & Stefan Schulz [1,2✉]

G protein-coupled receptors (GPCRs) are important signal transducers that are phosphorylated upon activation at intracellular serine and threonine residues. Although antibodies that specifically recognize the phosphorylation state of GPCRs have been available for many years, efficient immunolocalization of phosphorylated receptors in their tissues of origin has not been possible. Here, we show that phosphorylation of receptors is highly unstable during routine immunohistochemical procedures, requiring the use of appropriate phosphatase inhibitors particular during tissue perfusion, post-fixation, and cryoprotection but not during immunostaining of tissue sections. We provide proof of concept using phosphorylation state-specific μ-opioid receptor (MOP) and cannabinoid receptor 1 (CB1) antibodies. Indeed, three of four well-characterized phosphosite-specific MOP antibodies, including pS375-MOP, pT376-MOP, and pT379-MOP, showed robust neuronal immunostaining in brain and spinal cord sections of opioid-treated mice only after inclusion of phosphatase inhibitors. We then extended this approach to the CB1 receptor and demonstrated that one of three newly-generated phosphosite-specific CB1 antibodies, namely pS425-CB1, showed striking staining of fibers and varicosities in brain slices from cannabinoid-treated mice. Although subsequent experiments showed that phospho-CB1 immunostaining was less sensitive to phosphatases, we conclude that the use of phosphatase inhibitors should always be considered in the development of immunohistochemical procedures for new phosphosite-specific GPCR antibodies. In summary, we anticipate that this improved protocol will facilitate the widespread use of phosphorylation state-specific antibodies to monitor the activation of endogenous GPCRs under physiological and pharmacological conditions. Our approach may also prove useful to confirm target engagement of GPCR drug candidates in native tissues.

[1] Institut für Pharmakologie und Toxikologie, Universitätsklinikum Jena, Friedrich-Schiller-Universität Jena, Drackendorfer Straße 1, D-07747 Jena, Germany. [2] 7TM Antibodies GmbH, Hans-Knöll-Straße 6, D-07745 Jena, Germany. ✉email: stefan.schulz@med.uni-jena.de

A gonist-driven serine/threonine phosphorylation is a biologically and pharmacologically important process that primarily initiates desensitization and internalization of G protein-coupled receptors (GPCRs)[1–5]. Phosphorylation also increases the interaction of GPCRs with intracellular adapter proteins such as β-arrestins, which can trigger a second wave of signaling[1–5]. Thus, analysis of agonist-driven phosphorylation can provide valuable insights into the receptor activation state and ligand pharmacology. A widely used approach to analyze GPCR phosphorylation is the use of phosphosite-specific antibodies[6–9]. When available, such antibodies are valuable tools to elucidate the temporal dynamics of receptor phosphorylation, identify relevant kinases and phosphatases, and detect receptor activation using immunoblotting techniques[7, 10–15]. However, previous attempts using routine immunohistochemical approaches to unequivocally reveal agonist-induced phosphorylation of endogenous GPCRs in native tissues largely failed.

In intact cells, GPCR dephosphorylation is regulated in time and space, beginning immediately after receptor activation at the plasma membrane[16]. Indeed, phosphosite-specific antibodies in combination with siRNAs have led to the identification of distinct protein phosphatase 1 (PP1) and PP2 catalytic subunits as bona fide GPCR phosphatases[17–22]. For many receptors, dephosphorylation is complete within 10 to 30 minutes after agonist washout[17, 18, 21]. A notable exception is the agonist-induced phosphorylation of S341/S343 at the SST2 somatostatin receptor, which persists for considerably longer periods[21]. Consequently, activated SST2 receptors could be successfully localized using pS341/pS343-SST2 antibodies under routine immunohistochemical conditions[23]. However, this approach could not easily be reproduced for other SST2 sites or other GPCRs.

The ability to visualize activated and phosphorylated GPCRs in their tissues of origin would provide important clues to the physiological and pharmacological regulation of receptor activation. In particular, it would allow for distinction between currently-activated and resting GPCR populations in the context of particular physiological or behavioral conditions. We were therefore motivated to develop and validate immunohistochemical fixation and staining procedures that can be universally applied to prototypical GPCRs. Our improved protocol will facilitate the widespread application of phosphosite-specific antibodies as GPCR activation sensors in academic and pharmaceutical research.

## Results

Dephosphorylation of GPCRs occurs rapidly in intact cells and is mediated by the protein phosphatases PP1 and PP2[16]. Serine/threonine phosphatases are known to be activated during cell lysis and tissue fixation procedures. We therefore tested the inclusion of appropriate protein phosphatase inhibitors (PPIs) during tissue fixation and immunohistochemical staining. For this purpose, mice were treated with either methadone or saline, perfusion fixed, and slices of brain and spinal cord were stained with phosphosite-specific μ-opioid receptor (MOP) or phosphorylation-independent np-MOP antibodies. As expected, staining using the np-MOP antibody was present in the superficial layers of the spinal cord, regardless of drug treatment or the presence of phosphatase inhibitors (Fig. 1). In contrast, immunostaining for pT379-MOP was only detected in methadone-treated animals when a cocktail of appropriate phosphatase inhibitors was present during both fixation and staining procedures (Fig. 1). Four unique serine and threonine residues are phosphorylated in MOP in response to full agonists[8, 15]. Phosphosite-specific antibodies have been generated for all of these sites, which work equally well in immunoblot

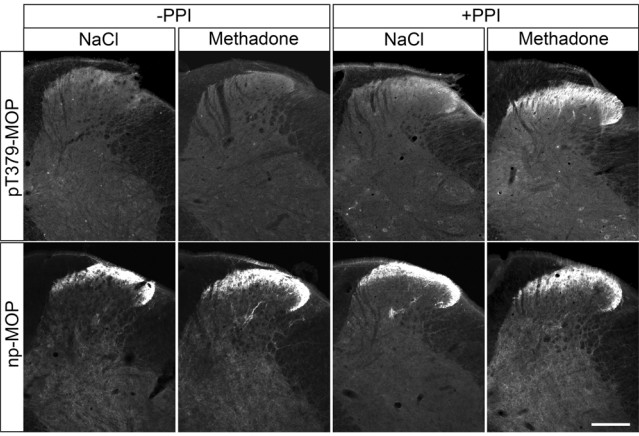

**Fig. 1 Comparison of phospho-MOP immunohistochemistry in the presence or absence of protein phosphatase inhibitors.** Animals were either treated with methadone or saline for 30 min, transcardially perfused, fixed and stained in the presence (+) or absence (-) of protein phosphatase inhibitors (PPI). Shown are confocal images of coronal sections of the spinal cord stained with pT379-MOP or np-MOP antibody. Note that PPIs need to be present during both fixation and staining procedures to obtain agonist-induced phospho-MOP immunostaining. Scale bar = 250 μm.

applications[7, 24, 25]. We found that three out of four, namely pS375-MOP, pT376-MOP and pT379-MOP, but not pT370-MOP, are also well suited for immunohistochemical staining of phosphorylated MOP under these conditions (Fig. 2 upper panel). Neither p-MOP nor np-MOP staining was apparent in MOP knockout mice (Fig. 2 lower panel). To further substantiate the specific binding of phospho-MOP antibodies, we performed peptide neutralization controls. When pS375-MOP or pT379-MOP antibodies were incubated with an excess of their respective immunizing peptides containing S375 or T379 in phosphorylated form, immunostaining was completely absent in the superficial layers of the spinal cord (Fig. 3). In contrast, immunostaining was virtually unaffected by the addition of the corresponding non-phosphorylated peptides, indicating unequivocal detection of agonist-dependent S375 and T379 phosphorylation of MOP in tissues (Fig. 3).

The antagonist naloxone drives MOP into an inactive state, preventing agonist-induced phosphorylation. When mice were treated with the selective MOP agonist fentanyl, we found the typical staining pattern for pS375-MOP, pT376-MOP and pT379-MOP as evidence of MOP activation. In contrast, this staining was not detected in animals injected with naloxone immediately prior to fentanyl challenge, indicating that MOP activation was blocked by the antagonist (Fig. 4). Collectively, our results support the notion that agonist-induced MOP phosphorylation is specifically detected in brain sections under these conditions. Notably, agonist-driven changes in phospho-immunofluorescence staining of MOP were robust enough to allow for quantification even in relatively small groups of animals (Supplemental Fig. S1). Given the considerable costs involved in using PPIs, we then tested whether their presence is necessary during both fixation and staining procedures. The results depicted in Fig. 5, clearly revealed that inclusion of PPIs during tissues fixation is an absolute requirement. They also show that after fixation, it is not obligatory to add PPIs to each solution for successful immunostaining (Fig. 5).

In an effort to perform dual immunofluorescence labeling, we combined the rabbit pT376-MOP antibody (depicted in green, Fig. 6) with guinea pig np-MOP antibody (depicted in magenta, Fig. 6). This procedure allows to discern populations of non-

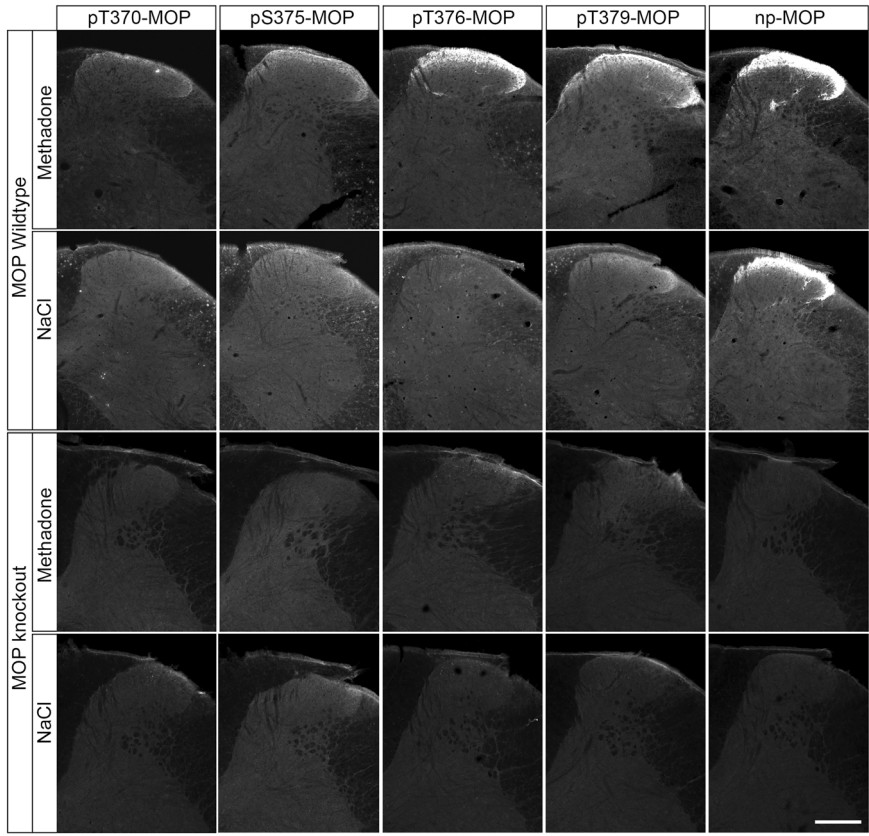

**Fig. 2 Immunohistochemical staining of agonist-induced MOP phosphorylation in the mouse spinal cord.** Animals were treated with methadone or saline for 30 min, transcardially perfused, fixed and stained in the presence of PPIs. Shown are confocal images of coronal sections of the spinal cord stained with phosphosite-specific antibodies pT370-MOP, pS375-MOP, pT376-MOP and pT379-MOP, or phosphorylation independent np-MOP antibody. Note that pS375-MOP, pT376-MOP and pT379-MOP, but not pT370-MOP, revealed an agonist-dependent MOP phosphorylation immunostaining in a pattern closely resembling that of np-MOP immunostaining. Scale bar = 250 μm.

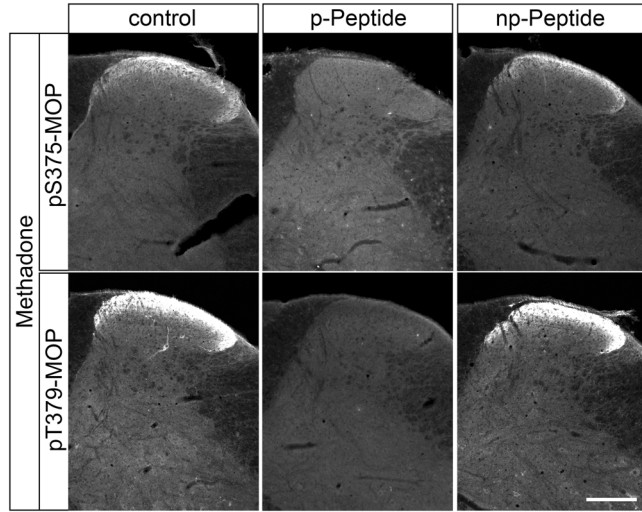

**Fig. 3 Peptide neutralization of agonist-induced MOP phosphorylation staining in the mouse spinal cord.** Animals were treated with methadone for 30 min, transcardially perfused, fixed and stained in the presence of PPIs. Shown are confocal images of coronal sections of the spinal cord stained with phosphosite-specific antibodies pS375-MOP or pT379-MOP in the presence or absence of their immunizing peptides containing S375 or T379 in phosphorylated form (p-Peptide) or the corresponding non-phosphorylated peptide (np-Peptide). Note that phospho-MOP immunostaining was completely neutralized by excess of phosphorylated peptide but not of non-phosphorylated peptide. Scale bar = 250 μm.

phosphorylated MOP from activated and phosphorylated MOP, which dramatically increases in the superficial layer of the spinal cord after administration of fentanyl or methadone (depicted in white in overlay images, Fig. 6). Next, we compared the immunostaining patterns obtained for phospho-MOP with total MOP. While there was good overlap in some brain regions, including habenula, fasciculus retroflex (Fig. 7), unexpected low or even absent pS375-MOP, pT376-MOP, nor pT379-MOP immunostaining was observed in the nucleus accumbens and caudate putamen, despite high levels of MOP expression (Fig. 7). These results suggest that region-specific MOP activation and phosphorylation patterns exist, which can now be visualized for the first time.

The final set of experiments was designed to extend this new approach to another prototypical GPCR namely the CB1 cannabinoid receptor. To this end, three newly-generated phosphosite-specific CB1 antibodies including pS425-CB1, pS429-CB1 and pS441-CB1 were tested under the new assay conditions. In fact, the pS425-CB1 antibody produced staining in hippocampus and cortex closely resembling that of the np-CB1 antibody only when mice were treated with CP-55940[26] (Fig. 8). Notably, pS425-CB1 immunostaining was much less prominent in untreated animals but not absent (Fig. 8). Staining in both untreated and cannabinoid-treated animals was completely abolished when the p425-CB1 antibody was neutralized with its immunizing peptide (Fig. 8). Prior administration of the antagonist AM251 reduced CP-55940-induced phospho-CB1 immunostaining to that seen in untreated mice (Fig. 8). To corroborate these findings using an independent biochemical

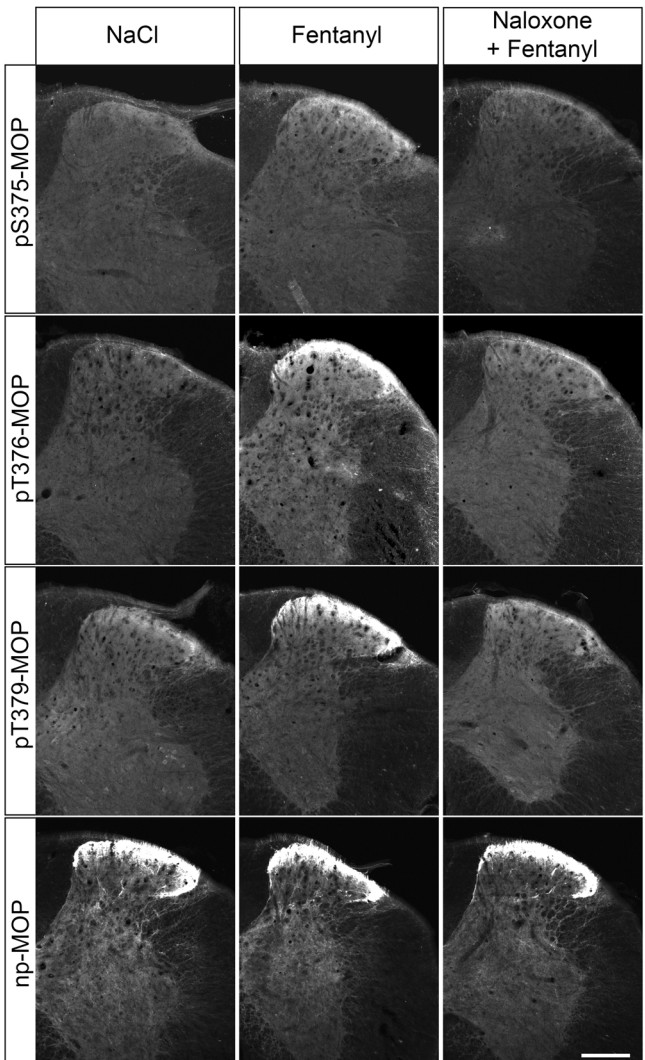

**Fig. 4 Antagonist block of phospho-MOP immunostaining.** Animals were either treated with saline or fentanyl for 15 min. Where indicated, animals were pretreated with naloxone for 10 min followed by 15-min fentanyl treatment. Animals were then transcardially perfused, fixed and stained in the presence of PPIs. Shown are confocal images of coronal sections of the spinal cord stained with phosphosite-specific antibodies pS375-MOP, pT376-MOP or pT379-MOP or phosphorylation independent np-MOP antibody. Note that agonist-induced phospho-MOP immunostaining is diminished by antagonist treatment. Scale bar = 250 μm.

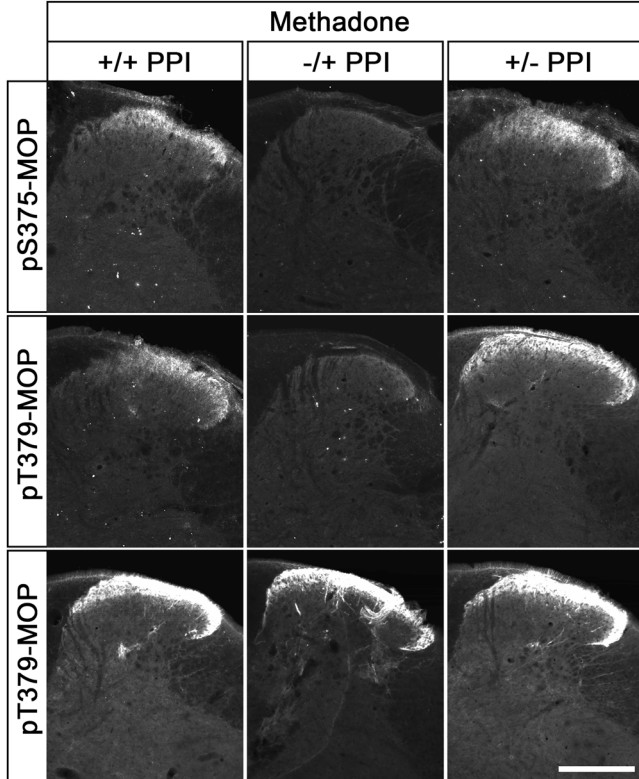

**Fig. 5 Comparison of phospho-MOP detection after staining in the presence or absence of PPIs.** Animals were treated with methadone for 30 min, transcardially perfused and fixed in the presence (+) or absence (-) of PPIs and also stained in the presence (+) or absence (-) of PPIs. Shown are confocal images of coronal sections of the spinal cord stained with pS375-MOP, pT379-MOP or np-MOP antibody. Note that PPIs need to be present during the whole fixation procedures but not during the staining protocol to obtain agonist-induced phospho-MOP immunostaining. Scale bar = 250 μm.

approach, groups of three animals each were treated with either vehicle or CP55940. CB1 receptors were then immunoprecipitated using a goat np-CB1 antibody and CB1 phosphorylation was determined using a rabbit pS425-CB1 antibody (Fig. 9). To confirm equal loading, the blot was stripped and total CB1 was detected using a rabbit np-CB1 antibody (Fig. 9). The results depicted in Fig. 9, reveal the existence of basal pS425 phosphorylation of CB1 that is strongly increased after administration of the high-efficacy agonist CP55940 (Fig. 9). When CB1 immunoprecipitates were subjected to λ-phosphatase digestion, both basal and agonist-driven CB1 phosphorylation were completely diminished, whereas total CB1 was still detectable. Collectively, these findings support the conclusion that CB1 phosphorylation can be specifically detected. We then tested whether the presence of PPIs during tissues fixation is an absolute requirement for successful pS425-CB1 immunostaining. Surprisingly, we found that CB1 phosphorylation appears to be much less sensitive to in situ phosphatase digestion than MOP phosphorylation (Supplemental Fig. S2). Finally, we used a combination of goat np-CB1 and rabbit pS425-CB1 antibodies to perform dual immunofluorescence labeling. At higher magnification it became apparent that the p425-CB1 antibody exclusively labeled fibers and terminals similar to that seen with the np-CB1 antibody (Fig. 10). Interestingly, overlay images revealed that in CP-55940-treated mice activated and phosphorylated CB1 receptors were densely concentrated in varicosities (depicted in white, Fig. 10), whereas CB1 was detected in fibers appears to reside in the non-phosphorylated state (depicted in magenta, Fig. 10).

## Discussion

GPCRs are privileged targets for small molecule drugs in almost all therapeutic areas[27]. However, to date, the visualization of therapeutic drug effects on endogenously expressed GPCRs in native tissues has remained a major challenge. Activation of GPCRs by their endogenous ligands or exogenous agonists results in conformational changes that are recognized by a family of kinases termed G protein-coupled receptor kinases (GRKs)[1–5]. The unique ability of GRKs to recognize activated receptors results in agonist-dependent phosphorylation at intracellular serine and threonine residues[1–5]. Thus, analysis of agonist-driven phosphorylation of GPCRs can provide valuable insights into receptor activation. Antibodies that specifically recognize the GPCR phosphorylation state have been available in some cases

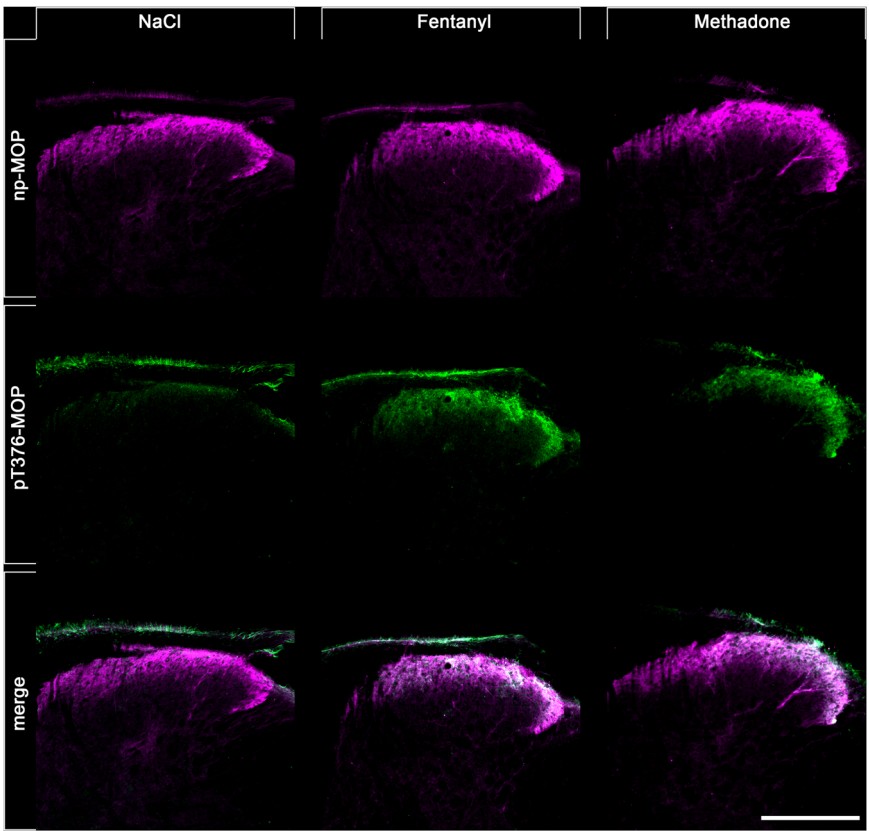

**Fig. 6 Double staining of agonist-induced p-MOP and np-MOP in mouse brain.** Animals were either treated with NaCl (30 min), Fentanyl (15 min) or Methadone (30 min). Animals were then transcardially perfused, fixed and stained. Shown are confocal images of coronal sections of the spinal cord stained with phosphosite-specific antibody pT376-MOP (green) and phosphorylation independent np-MOP antibody (magenta). Note that there is a strong overlap of np-MOP and pT376-MOP in the merged panels (white) in agonist treated animals compared to NaCl treated mice. Scale bar = 250 μm.

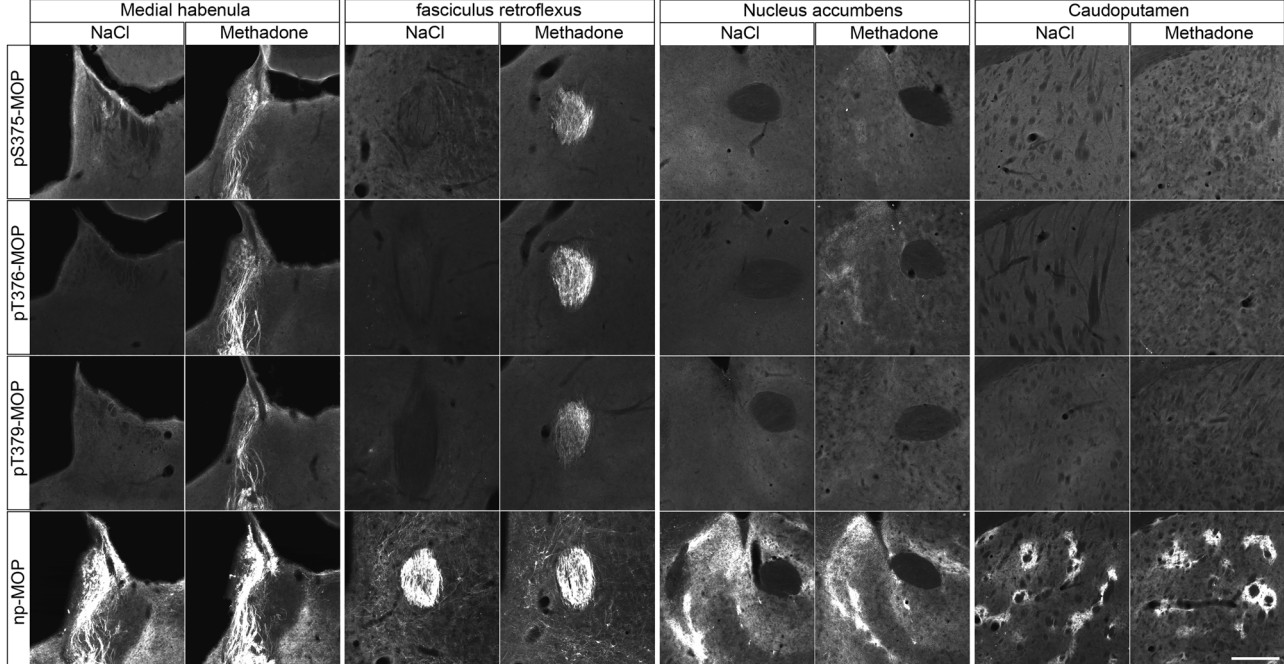

**Fig. 7 Agonist-induced phospho-MOP immunostaining in mouse brain.** Animals were treated with methadone for 30 min, transcardially perfused, fixed and stained in the presence of PPIs. Shown are confocal images of coronal brain sections stained with phosphosite-specific antibodies pS375-MOP, pT376-MOP or pT379-MOP or phosphorylation independent np-MOP antibody. Note that all three phosphosite-specific antibodies detected agonist-induced MOP phosphorylation in the medial habenula and fasciculus retroflexus but not in nucleus accumbens or caudate putamen, brain regions known to be rich in MOP receptors as shown by np-MOP antibody staining. Scale bar = 250 μm.

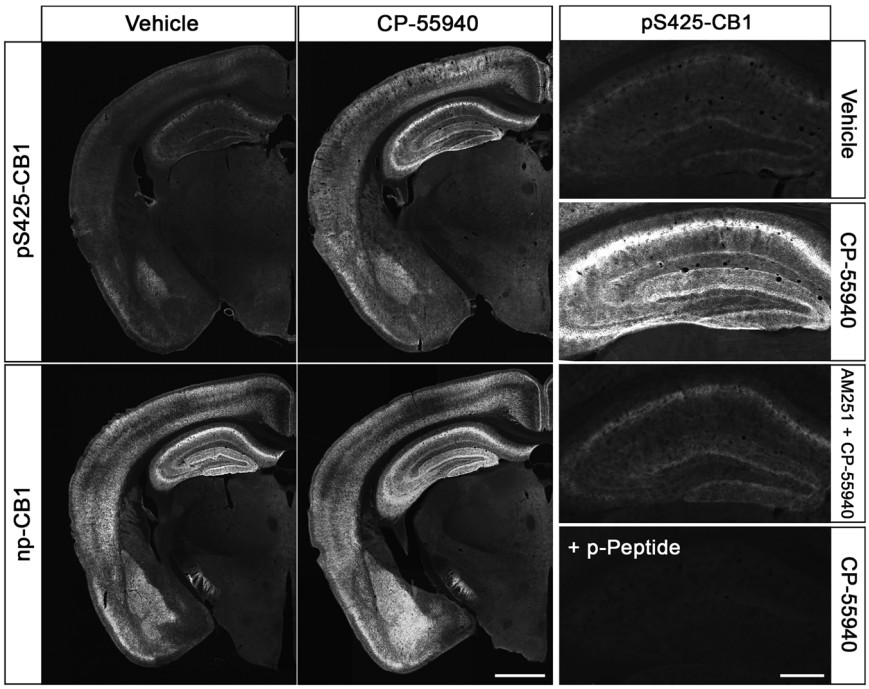

**Fig. 8 Agonist-induced phospho-CB1 immunostaining in mouse brain.** Animals were either treated with Vehicle or the CB1-agonist CP-55940 for 30 min. Where indicated, animals were pretreated with the CB1-antagonistAM251 for 45 min followed by 30 min CP-55940 treatment. Animals were then transcardially perfused, fixed and stained in the presence of PPIs. Shown are confocal images of coronal brain sections stained with phosphosite-specific antibody pS425-CB1, in the presence or absence of the p-Peptide containing S425 in phosphorylated form, or phosphorylation independent np-CB1 antibody. Note that agonist-induced phopho-CB1 immunostaining is diminished by antagonist treatment. Note that phospho-CB1 immunostaining was completely neutralized by excess of phosphorylated peptide. Scale bar = 1000 μm and 250 μm.

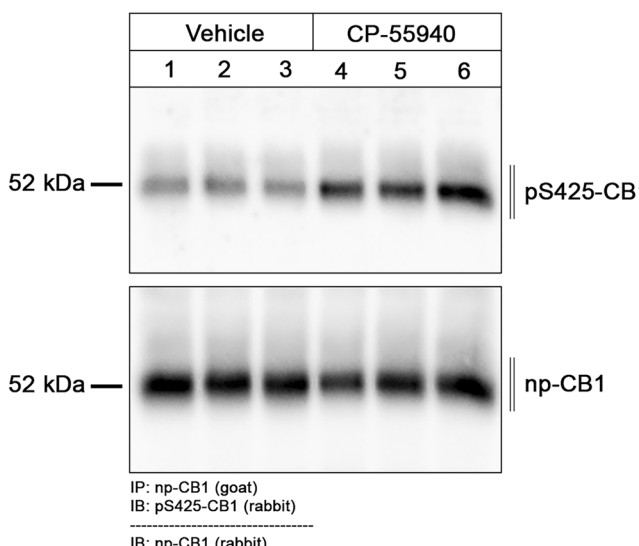

**Fig. 9 Immunoblotting of phosphorylated CB1 in agonist treated mice.** Six different wildtype animals were used. Number 1 to 3 were treated with vehicle and Number 4 to 6 with CP-55940 for 30 min. Organs were removed rapidly and frozen in liquid nitrogen After tissue preparations, CB1 receptors were extracted using goat np-CB1 bound to protein A beads. After blotting, the membranes were treated with the phosphosite-specific rabbit antibody pS425-CB1, stripped (dashed line) and reprobed with rabbit phosphorylation-independent anti-CB1. Note that there is some CB1-phosphorylation constitutive signal in vehicle-treated mice. Molecular mass on the left in kDa.

over the last 20 years. Indeed, phosphosite-specific GPCR antibodies have proven useful to detect receptor activation and to profile the pharmacological properties of new ligands[12–15]. To date, however, their utility has been largely limited to immunoblotting approaches using heterologous receptor expression systems, while detection of phosphorylated GPCRs from tissue lysates has been mostly unsuccessful, due to their low abundance in vivo.

Here, we have systematically developed an improved immunohistochemical staining procedure that allows for visualization of opioid drug effects in the mouse brain in vivo. We also show that this approach can be easily extended to other prototypical GPCRs. Key to this new protocol was the observation that receptor phosphorylation is highly unstable during routine immunohistochemistry and that for many GPCRs inclusion of protein phosphatase inhibitors may be an absolute requirement during perfusion, post-fixation and cryoprotection of tissues. Several lines of evidence suggest that our approach facilitates the localization of bona fide phosphorylated GPCRs. First, phospho-GPCR immunostaining strongly increased when animals had been treated with agonist. Conversely, immunostaining was not detected when agonist action was blocked by antagonist. Second, robust phospho-GPCR immunostaining was only apparent when appropriate protein phosphatase inhibitors were included in all steps of fixation and staining procedures. Third, phospho-GPCR immunostaining was completely neutralized by preincubation with cognate phosphorylated peptides but not with the corresponding non-phosphorylated peptides.

In the present study, we had access to four different phospho-MOP and three different phospho-CB1 antibodies which had previously been validated by immunoblotting techniques. We

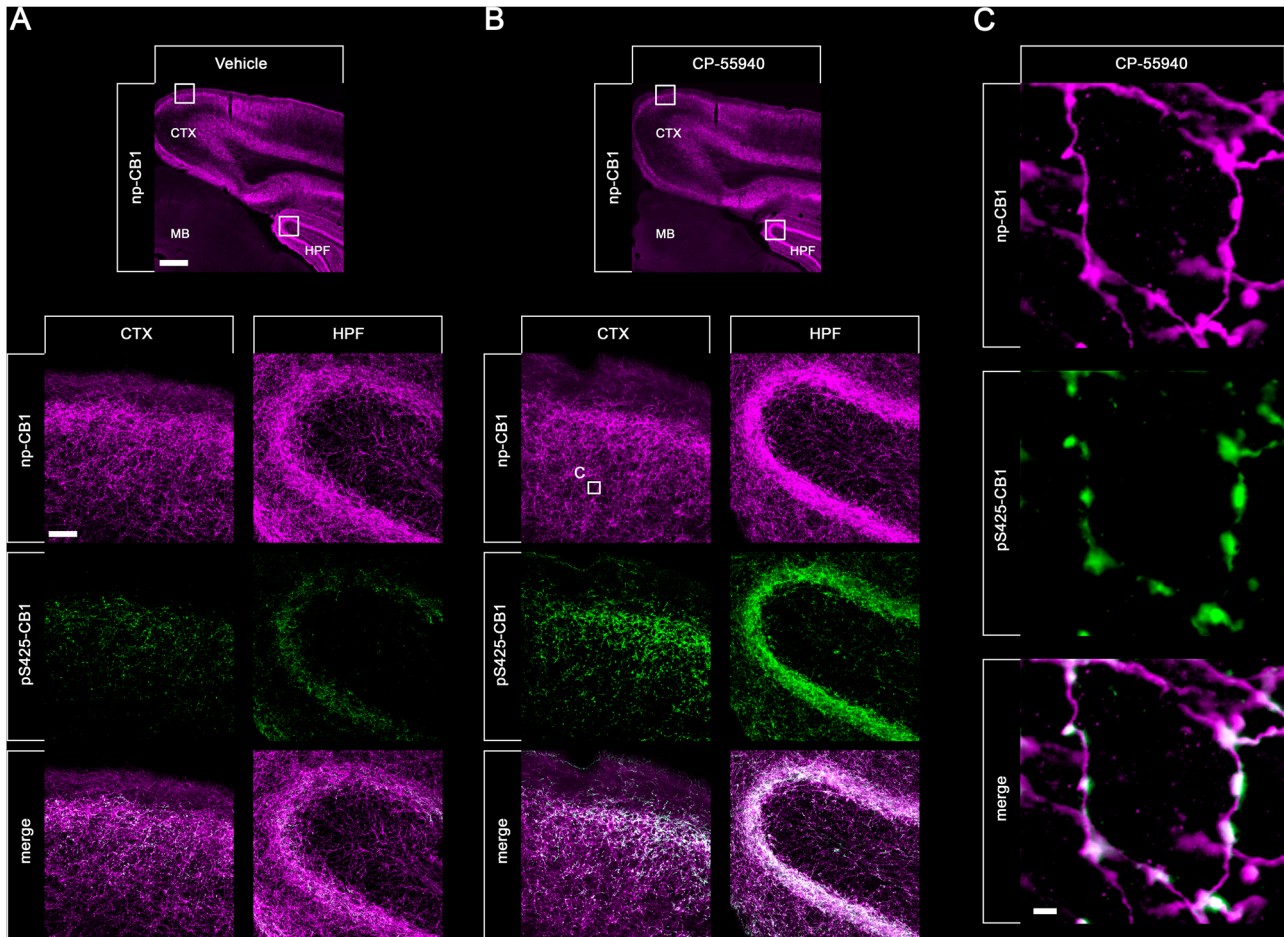

**Fig. 10 Double staining of agonist-induced p-CB1 and np-CB1 in mouse brain.** Animals were either treated with Vehicle (**A**) or CP-55940 (**B**, **C**) for 30 min. Animals were then transcardially perfused, fixed and stained. Shown are confocal images of coronal brain sections of cortex (CTX) and hippocampal formation (HPF) stained with phosphosite-specific antibody pS425-CB1 (green) and phosphorylation independent np-CB1 antibody (magenta). Note that there is a strong overlap of np-CB1 and pS425-CB1 in the merged panels (white) in CP-55940 treated animals (**B**) compared to vehicle treated mice. Note that in cortical neurons, only CB1 located in varicosities undergoes agonist-induced phosphorylation. Scale bar = 1000 μm, 100 μm and 2 μm.

observed that three out of four antibodies for MOP and one out of three antibodies for CB1 produced specific phospho-immunostaining in tissues suggesting that each phosphosite-specific GPCR antibody has to be validated individually for immunohistochemical applications. Moreover, we observed that phospho-MOP staining was critically dependent on the presence of PPIs, whereas CB1 phosphorylation appeared to be more stable in tissues. Nevertheless, we conclude that use of PPIs should always be considered in the development of immunohistochemical procedures for new phosphosite-specific GPCR antibodies.

It is well known that endogenous GPCRs are notoriously difficult to detect in tissues by immunohistochemical staining. Here, we not only establish a protocol for in situ visualization of activated and phosphorylated GPCRs. We also demonstrate that phospho-immunostaining can disclose previously unappreciated functional information not seen with phosphorylation-independent GPCR antibodies. One example is the striking mismatch between total and phospho-immunostaining for MOP in caudate putamen. This mismatch cannot be explained be low detection level of phospho-MOP staining as many other brain regions with comparable total MOP revealed prominent phospho-MOP. The most likely explanation for the relative lack of phospho-MOP staining in caudate putamen is the low or absent expression of relevant GRK family members (Allen Brain Atlas, https://atlas.brain-map.org). Notably,

lack of MOP phosphorylation in some brain regions relative to others could have profound consequences on MOP desensitization and may explain well-known differences in tolerance development to different opioid effects. Another example is the different subcellular localization of phosphorylated and non-phosphorylated CB1 receptors. While total CB1 appears to be evenly distributed along fibers and varicosities, phosphorylated CB1 is particular enriched in terminals indicating compartmentalization of GRK-mediated phosphorylation.

In summary, we provide a proof of concept for using phosphorylation state-specific antibodies as sensors to elucidate GPCR activation in native tissues. Our approach not only facilitates the visualization of the effects of agonists and antagonists in pre-clinical animal studies. It also allows for elucidation of different cellular and subcellular activation patterns in response to external stimuli or drug administration. Therefore, phosphorylation state-specific antibodies are a new class of markers for GPCR activation that are likely to find widespread application in academic and pharmaceutical research.

## Methods
### Antibodies
*Primary antibodies.* The phosphorylation state-specific rabbit MOP antibodies pT370-MOP (7TM0319B), pS375-MOP (7TM0319C), pT376-MOP (7TM0319D),

pT379-MOP (7TM0319E), the phosphorylation state-specific rabbit CB1 antibody pS425-CB1 (7TM0056A) as well as the phosphorylation-independent rabbit antibody np-MOP (7TM0319N) and phosphorylation-independent guinea pig antibody np-MOP (7TM0319N-GP) were provided by 7TM Antibodies (www.7tmantibodies.com). In addition, we used rabbit np-MOP from abcam (ab134054), rabbit np-CB1 from Cayman (1000659) and goat np-CB1 from Frontier Institute (MFSR100610). Additional information in Supplemental Table 1.

*Secondary antibodies.* We used a biotinylated AB from Jackson Immuno Research (711-065-152, anti-rabbit) and the respective streptavidine conjugates from ThermoFisher (S21381, Alexa Fluor™ 555 conjugate) and (S11223, Alexa Fluor™ 488 conjugate) for the amplified staining procedure. For the direct stainings we used Cy™3 (706-165-148, anti-guinea pig) and Cy™3 (705-165-147, anti-goat), both obtained from Jackson ImmunoResearch.

**Animals.** Mice (C57BL/6 J obtained from JAX™) were housed 2–5 per cage under a 12-hr light-dark cycle with ad libitum access to food and water. All animal experiments were approved by Thuringian state authorities and performed in accordance with European Commission regulations for the care and use of laboratory animals. Our study is reported in accordance with ARRIVE guidelines[28]. In all experiments, male and female mice aged 8–30 weeks between 25 and 35 g body weight were used.

**Drugs and routes of administration.** All drugs were freshly prepared prior to use and were injected subcutaneously (s.c.) or intraperitoneally (i.p.) in unanaesthetized mice at a volume of 10 μl/g bodyweight. Opioid drugs were diluted in 0.9% (w/v) saline and cannabinoid drugs in vehicle solution, containing 13.2% ethanol and 0.25% Tween80 in 0.9% saline, for injections. Drugs were obtained and used as follows: fentanyl citrate (0.3 mg/kg for 15 min; s.c.) (B. Braun 06900650), levomethadone hydrochloride (15 mg/kg for 30 min; s.c.) (Sanofi-Aventis 07480196), naloxone (2 mg/kg for 25 min; s.c.) (Ratiopharm 04788930), CP-55940 (0.75 mg/kg for 30 min; i.p.) (Sigma-Aldrich C1112) and AM251 (3 mg/kg for 45 min; i.p.) (MedChemExpress 183232-66-8). A total of 35 mice ($n = 4$–6 per treatment condition) were used.

**Immunohistochemistry.** Mice were deeply anaesthetized with isoflurane (CP-Pharma 4001404) and then subjected to a transcardial perfusion with calcium-free Tyrode's solution containing protein phosphatase-inhibitors (+PPIs) (1 tablet PhosSTOP per 10 ml) (Roche 04906845001) followed by Zamboni's fixative containing 4% paraformaldehyde and 0.2% picric acid in 0.1 M phosphate buffer pH 7.4 +PPIs. Brains and spinal cords were rapidly dissected and postfixed in the same fixative for 4 h at room temperature. The tissue was cryoprotected by immersion in 10% sucrose +PPIs followed by 30% sucrose +PPIs for 48 h at 4 °C before sectioning using a freezing microtome. Tissue was cut into 40 μm sections. Free-floating sections were washed in PBS with or without (+/-) PPIs (depending on the experiment and investigated GPCR) and incubated in methanol containing 0.3% $H_2O_2$ for 30 min. After washing in PBS-T (PBS + 0,3% Tween®20), the sections were blocked in PBS containing 0.3% Triton X-100, +/- PPIs and 10% NGS for 2 h. Subsequently, the sections were incubated with primary antibody in PBS containing 0.3% Triton X-100, 2% NGS + /- PPI overnight at 4 °C. Where indicated, primary antibodies were preincubated with 1 μg/ml of their cognate phospho-peptide or the corresponding non-phospho-peptide for 1 h at room temperature. When necessary, staining of primary antibody was detected using the biotin amplification procedure as described[29, 30]. Briefly, tissue sections were transferred to biotinylated donkey anti-rabbit IgG (1:300 in PBS containing 0.3% Triton X-100, +/-PPIs and 10% NGS) for 2 h, washed in PBS-T + /-s PPIs and then incubated in AB solution (reagents from the Vector ABC kit; 25 μl A and 25 μl B in 10 ml PBS + 0.3% Triton X-100 + /- PPIs) for 60 min, washed again in PBS-T + /- PPIs and transferred to biotinylated tyramine (BT) solution (BT was prepared as described by Adams[31], 5 μl BT + 0.01% $H_2O_2$ in 1 ml PBS + 0.3% Triton X-100 and + /- PPIs) for 20 min, followed by a final incubation step in streptavidin-AlexaFluor555 conjugate 1:400 in PBS + 0.3% Triton X-100 containing 10% NDS +/− PPI overnight at 4 °C. For double-staining procedures, the primary np-AB was incubated overnight as indicated and after several washing steps on the following day conjugated to the secondary AB, again overnight. Sections were then mounted onto SuperFrost Plus glass slides (ThermoFisher 15438060) and cover slipped with Eukitt (ORSAtec). Specimens were examined using a Zeiss LSM 900 laser scanning confocal microscope equipped with ZEN software for image analysis.

**Western Blot and immunoprecipitation.** Wildtyp mice were treated with agonist or vehicle. Mice were anesthetized with isoflurane, killed by cervical dislocation, and brains were quickly dissected, excluding the cerebellum. Brain samples were immediately frozen in liquid nitrogen. Brains were transferred to ice-cold detergent buffer (50 mM Tris-HCl, pH 7.4, 150 mM NaCl, 5 mM EDTA, 1% Nonidet P-40, 0.5% sodium deoxycholate, 0.1% sodium dodecyl sulfate (SDS), containing protease and phosphatase inhibitors), homogenized, and centrifuged at $14,000 \times g$ for 30 min at 4 °C. The supernatant was then immunoprecipitated with the polyclonal goat np-CB1 antibody bound to protein A-agarose beads (20333, ThermoScientific,

Germany) for 90 min at 4 °C. Proteins were eluted from the beads with SDS sample buffer for 25 min at 43 °C and then resolved on 8% SDS-polyacrylamide gels. After electroblotting, membranes were incubated with pS425-MOP antibody, followed by detection using a chemiluminescence detection system. Blots were subsequently stripped and incubated again with the rabbit np-CB1 antibody to confirm equal loading of the gels.

**Reporting summary.** Further information on research design is available in the Nature Portfolio Reporting Summary linked to this article.

## Data availability

All data supporting the findings of this study are available within the article and its supplementary information files (uncropped blots in Supplementary Fig. 3 and source data for graphs in Supplementary Data 1). Additional information, relevant data and unique biological materials will be available from the corresponding author upon reasonable request. Source data are provided with this paper.

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

## Acknowledgements
We want to thank Svetlana Würl and Monique Brendel for expert technical assistance. This work was supported by the European Regional Development Fund (EFRE) and the Free State of Thuringia (grant: 2020 FE 0146) to 7TM Antibodies GmbH.

## Author contributions
S.S. conceived and initiated the project and designed all experiments with S.F. S.F. performed and analyzed immunohistochemistry. A.K. and V.S. performed immuno- blotting assay. F.N. affinity-purified all phosphosite-specific antibodies used in this study. The manuscript was written and revised by S.S. and S.F. with input from other authors.

## Funding

## Competing interests
S.S. is the founder and scientific advisor of 7TM Antibodies GmbH, Jena, Germany. F.N. is an employee of 7TM Antibodies. All other authors declare no competing interests.
