## [Peer Review File · Communications Biology]

Reviewers' comments:

Reviewer #1 (Remarks to the Author):

This manuscript by Fritzwanker and colleagues details the inclusion of phosphatase inhibitors during processing for immunohistochemistry as a means to retain phosphates on activated GPCRs and thereby allow for the detection of phosphorylated GPCRs in tissues sections. It is an obvious and relatively straightforward advance and the manuscript is primarily methodical in scope. It is well written and the data are believable and well controlled (good use of receptor knockout mice and blocking peptides to show specificity).

However, I am not convinced that there is enough biology in this manuscript to warrant publication in a journal like Communications Biology and its growing reputation/impact (IF ~6).

I would expect that the application of the inhibitors and showing enhanced staining would form a part of a figure in a paper that is using the information to advance our understanding of the biology of the receptors and their contribution to physiology and pathophysiology.

The manuscript is professionally presented and so there are no major comments except the idea in the final paragraph of the discussion that this approach might be used on human tissues. Given the approach requires infusion with phosphatase inhibitors on a still living organism, perhaps some expansion on this idea is required when discussing application to human tissues. Also, has the approach been used in paraffin embedded tissue sections? These are widely used and provide better cellular resolution?

Reviewer #2 (Remarks to the Author):

This is an important and well-presented study which addresses the challenge of analyzing protein phosphorylation in fixed tissue using phosphospecific antibodies. The work is carried out by a group with high expertise in this general area, and the manuscript provides clear and well-documented guidelines for achieving high detection sensitivity by avoiding artifactual dephosphorylation during and after fixation. I was surprised by what a big difference can be made with routine inclusion of inhibitors as the authors recommend. I think this work will make an important contribution.

I have no significant specific criticisms and believe that the manuscript is suitable in its present form.

Reviewer #3 (Remarks to the Author):

In this manuscript, the authors test whether adding phosphatase inhibitors (PPIs) during the perfusion and IHC process improves the effectiveness of phospho-specific antibodies. Here, they focus on the mu-opioid receptor, and show that addition of PPIs enhances several phospho-antibodies while not affecting non-phospho staining. A KO, blocking peptide, and antagonist controls are critically present. They go on to show mu staining in several brain areas and suggest there may be phospho-differences, perhaps pointing to functional utility of these antibodies. Finally, they demonstrate that the CB1R can also be visualized using this technique.

Generally, the manuscript is well-written, and the results tend to support the idea that PPIs can have a beneficial effect on phospho-GPCR IHC in tissue sections. There are several theoretical, methodological, and analytic concerns that should be addressed prior to publication, detailed below.

1) Calling phosphorylation the "most important" modification is subjective, and this should probably be reworded to something like "very important". It may also be worth a phrase on why it is a very important process.

2) One prevailing theme throughout the manuscript is that PPIs are required both during perfusion and during the subsequent IHC process. However, as far as I can tell, this has not been explicitly tested/reported. There is no condition where PPIs are only included during perfusion/post-fixation and/or only during the IHC process. Therefore, the authors cannot conclude that they are required for both. If this data is available, then it should certainly be included. Note that the authors refer to supplemental data (line 273) but no supplement was included in the pdf I received nor referenced in the body of the MS.

This is important because on a theoretical level, it seems very difficult to understand why PPIs would be needed post-fixation, at a time when there should be no PP enzymatic activity. Phosphatases should be cross-linked and inactivated as part of the fixation process, and certainly once the brain is post-fixed and sliced. Practically, while PPIs during fixation is a relatively easy process, making sure there are PPIs in every solution during IHC could be cumbersome and potentially superfluous.

Therefore, without further data, the language needs to be changed to not have "required" (eg line 131), because it could be just the perfusion and post-fix period that is crucial.

3) Many of the figures in this paper use only 1 representative image to make the authors' conclusion, despite 4-6 animals being used per treatment. It is thus unclear what kind of variance in intensity/staining there is with this method. Because there should be multiple tissue samples/stains available per treatment, some type of fluorescence intensity quantification should be used to help show the type of variance observed with this method. This is also important because the authors use both sexes and a very large age range (8-30 weeks) which could add variability.

4) The authors nicely demonstrate that this method works for a CB1R p-antibody, using appropriate controls. However, there is no PPI (-) condition so it is hard to know whether PPIs actually benefited this experiment, or whether the pCB1 would work under normal IHC conditions. The PPI (-) needs to be included to demonstrate any benefit of the technique.

5) All of the phospho-antibodies in this study were provided by 7TM Antibodies, and 2 of the authors have relationships with the company (including the founder). While I have no explicit problem with this, it seems that at least 1 p-antibody from another company should be used to validate this approach. Otherwise it's unclear whether this really is a generalizable process for 'all' phospho-antibodies, or just for the antibodies from this company.

6) It's unusual that JAXTM Bl/6J mice were obtained from Charles River, because the terms "JAX" and "J" indicate Jackson Labs. The authors may double check their mouse source.

7) "also in human tissues" (line 143). This is presumptive, as no human tissue was examined here. Unless there is a rationale for this sentence, it should be removed, as this would also require large changes in the way humans are fixed/preserved. Also, the last sentence of the discussion should be modified as phospho-antibodies are not a "new" class of biomarkers, having been around at least 20 years as the authors note above in the discussion (lines 119-120).

Response to Reviewers

Black - Reviewer comments

Green – Author comments

Blue – new/edited text

Reviewer #1 (Remarks to the Author):

This manuscript by Fritzwanker and colleagues details the inclusion of phosphatase inhibitors during processing for immunohistochemistry as a means to retain phosphates on activated GPCRs and thereby allow for the detection of phosphorylated GPCRs in tissues sections. It is an obvious and relatively straightforward advance and the manuscript is primarily methodical in scope. It is well written and the data are believable and well controlled (good use of receptor knockout mice and blocking peptides to show specificity).

However, I am not convinced that there is enough biology in this manuscript to warrant publication in a journal like Communications Biology and its growing reputation/impact (IF ~6). I would expect that the application of the inhibitors and showing enhanced staining would form a part of a figure in a paper that is using the information to advance our understanding of the biology of the receptors and their contribution to physiology and pathophysiology.

We have added new data Figures 5, 6, 9, 10. Discussion line 199 reads now: “It is well known that endogenous GPCRs are notoriously difficult to detect in tissues by immunohistochemical staining. Here, we not only establish a protocol for in situ visualization of activated and phosphorylated GPCRs. We also demonstrate that phospho- immunostaining can disclose previously unappreciated functional information not seen with phosphorylation-independent GPCR antibodies. One example is the striking mismatch between total and phospho- immunostaining for MOP in caudate putamen. This mismatch cannot be explained by low detection level of phospho-MOP staining as many other brain regions with comparable total MOP revealed prominent phospho-MOP. The most likely explanation for the relative lack of phospho-MOP staining in caudate putamen is the low or absent expression of relevant GRK family members³². Notably, lack of MOP phosphorylation in some brain regions relative to others could have profound consequences on MOP desensitization and may explain well-known differences in tolerance development to different opioid effects. Another example is the different subcellular localization of phosphorylated and non-phosphorylated CB1 receptors. While total CB1 appears to be evenly distributed along fibers and varicosities phosphorylated CB1 is particular enriched in terminals indicating compartmentalization of GRK-mediated phosphorylation.”

The manuscript is professionally presented and so there are no major comments except the idea in the final paragraph of the discussion that this approach might we used on human tissues. Given the approach requires infusion with phosphatase inhibitors on a still living organism, perhaps some expansion on this idea is required when discussing application to human tissues. Also, has the approach been used in paraffin embedded tissue sections? These are widely used and provide better cellular resolution?

Reference to human tissues in the discussion has been deleted.

Reviewer #2 (Remarks to the Author):

This is an important and well-presented study which addresses the challenge of analyzing protein phosphorylation in fixed tissue using phosphospecific antibodies. The work is carried out by a group with high expertise in this general area, and the manuscript provides clear and well-documented guidelines for achieving high detection sensitivity by avoiding artifactual dephosphorylation during and after fixation. I was surprised by what a big difference can be made with routine inclusion of inhibitors as the authors recommend. I think this work will make an important contribution.

I have no significant specific criticisms and believe that the manuscript is suitable in its present form.

Reviewer #3 (Remarks to the Author):

In this manuscript, the authors test whether adding phosphatase inhibitors (PPIs) during the perfusion and IHC process improves the effectiveness of phospho-specific antibodies. Here, they focus on the mu-opioid receptor, and show that addition of PPIs enhances several phospho-antibodies while not affecting non-phospho staining. A KO, blocking peptide, and antagonist controls are critically present. They go on to show mu staining in several brain areas and suggest there may be phospho-differences, perhaps pointing to functional utility of these antibodies. Finally, they demonstrate that the CB1R can also be visualized using this technique.

Generally, the manuscript is well-written, and the results tend to support the idea that PPIs can have a beneficial effect on phospho-GPCR IHC in tissue sections. There are several theoretical, methodological, and analytic concerns that should be addressed prior to publication, detailed below.

1) Calling phosphorylation the “most important” modification is subjective, and this should probably be reworded to something like “very important”. It may also be worth a phrase on why it is a very important process.

Introduction line 45 has been changed as follows: “Agonist-driven serine/threonine phosphorylation is a biologically and pharmacologically important process that primarily initiates desensitization and internalization of G protein-coupled receptors (GPCRs)¹⁻⁵. Phosphorylation also increases the interaction of GPCRs with intracellular adapter proteins such as β -arrestins, which can trigger a second wave of signaling¹⁻⁵. Thus, analysis of agonist-driven phosphorylation can provide valuable insights into the receptor activation state and ligand pharmacology. “

2) One prevailing theme throughout the manuscript is that PPIs are required both during perfusion and during the subsequent IHC process. However, as far as I can tell, this has not

been explicitly tested/reported. There is no condition where PPIs are only included during perfusion/post-fixation and/or only during the IHC process. Therefore, the authors cannot conclude that they are required for both. If this data is available, then it should certainly be included. Note that the authors refer to supplemental data (line 273) but no supplement was included in the pdf I received nor referenced in the body of the MS.

This is important because on a theoretical level, it seems very difficult to understand why PPIs would be needed post-fixation, at a time when there should be no PP enzymatic activity. Phosphatases should be cross-linked and inactivated as part of the fixation process, and certainly once the brain is post-fixed and sliced. Practically, while PPIs during fixation is a relatively easy process, making sure there are PPIs in every solution during IHC could be cumbersome and potentially superfluous.

Therefore, without further data, the language needs to be changed to not have “required” (eg line 131), because it could be just the perfusion and post-fix period that is crucial.

We conducted the suggested experiments, added Figure 5 and revised the text line 111 accordingly: “Given the considerable costs involved in using PPIs, we then tested whether their presence is necessary during both fixation and staining procedures. The results depicted in Figure 5, clearly revealed that inclusion of PPIs during tissues fixation is an absolute requirement. They also show that after fixation, it is not obligatory to add PPIs to each solution for successful immunostaining (Fig. 5).”

3) Many of the figures in this paper use only 1 representative image to make the authors’ conclusion, despite 4-6 animals being used per treatment. It is thus unclear what kind of variance in intensity/staining there is with this method. Because there should be multiple tissue samples/stains available per treatment, some type of fluorescence intensity quantification should be used to help show the type of variance observed with this method. This is also important because the authors use both sexes and a very large age range (8-30 weeks) which could add variability.

We have added quantification in Supplemental Figure S2 and revised the text line 109 accordingly: “Notably, agonist-driven changes in phospho-immunofluorescence staining of MOP were robust enough to allow for quantification even in relatively small groups of animals (Supplemental Figure S1).”

4) The authors nicely demonstrate that this method works for a CB1R p-antibody, using appropriate controls. However, there is no PPI (-) condition so it is hard to know whether PPIs actually benefited this experiment, or whether the pCB1 would work under normal IHC conditions. The PPI (-) needs to be included to demonstrate any benefit of the technique.

We conducted the suggested experiment, added Supplemental Figure S2 and revised the text line 148 accordingly: “We then tested whether the presence of PPIs during tissues fixation is an absolute requirement for successful pS425-CB1 immunostaining. Surprisingly, we found that CB1 phosphorylation appears to be much less sensitive to in situ phosphatase digestion than MOP phosphorylation (Supplemental Fig. S2).”

5) All of the phospho-antibodies in this study were provided by 7TM Antibodies, and 2 of the authors have relationships with the company (including the founder). While I have no explicit problem with this, it seems that at least 1 p-antibody from another company should be used to validate this approach. Otherwise it's unclear whether this really is a generalizable process for 'all' phospho-antibodies, or just for the antibodies from this company.

In response to the Reviewer, we have tested pS375-MOP from CST and found this antibody not suitable for phospho-MOP IHC staining (although we found this antibody useful for immunoblotting in previous studies. Thus, it is not possible to predict the applicability of a given phospho-GPCR antibody for IHC staining based on immunoblotting results. To make this fact clear for the reader we have added the following paragraph at line 190: "In the present study, we had access to four different phospho-MOP and three different phospho-CB1 antibodies which had previously been validated by immunoblotting techniques. We observed that three out of four antibodies for MOP and one out of three antibodies for CB1 produced specific phospho-immunostaining in tissues suggesting that each phosphosite-specific GPCR antibody has to be validated individually for immunohistochemical applications. Moreover, we observed that phospho-MOP staining was critically dependent on the presence of PPIs, whereas CB1 phosphorylation appeared to be more stable in tissues. Nevertheless, we conclude that use of PPIs should always be considered in the development of immunohistochemical procedures for new phosphosite-specific GPCR antibodies."

6) It's unusual that JAXTM Bl/6J mice were obtained from Charles River, because the terms "JAX" and "J" indicate Jackson Labs. The authors may double check their mouse source.

Animal source has been corrected.

7) "also in human tissues" (line 143). This is presumptive, as no human tissue was examined here. Unless there is a rationale for this sentence, it should be removed, as this would also require large changes in the way humans are fixed/preserved. Also, the last sentence of the discussion should be modified as phospho-antibodies are not a "new" class of biomarkers, having been around at least 20 years as the authors note above in the discussion (lines 119-120).

Reference to human tissues in the discussion has been deleted and text at line 219 has been changed: "phosphorylation state-specific antibodies are a new class of biomarkers for GPCR activation"

REVIEWERS' COMMENTS:

Reviewer #1 (Remarks to the Author):

I have no further comments.

Reviewer #3 (Remarks to the Author):

The authors have done a commendable job of responding to each of my comments, with either text changes, or more importantly, critical experiments. The additional data more clearly demonstrates the benefits of the approach, and I think it will make a substantial impact in p-antibody protocols. I have no further issue, and believe it is now suitable for publication.